# Characterization of New Alpha Zero (α^0^) Thalassaemia Deletion (−−^GB^) among Malays in Malaysian Population

**DOI:** 10.3390/diagnostics13203286

**Published:** 2023-10-23

**Authors:** Norafiza Mohd Yasin, Faidatul Syazlin Abdul Hamid, Syahzuwan Hassan, Yuslina Mat Yusoff, Ermi Neiza Mohd Sahid, Ezalia Esa

**Affiliations:** Haematology Unit, Cancer Research Center, Institute for Medical Research, National Institutes of Health, Ministry of Health, Shah Alam 40170, Selangor, Malaysiasyah606@gmail.com (S.H.); yuslina.my@moh.gov.my (Y.M.Y.); ezalia@moh.gov.my (E.E.)

**Keywords:** GB deletion, alpha zero thalassaemia, deletional characterization, unique mutation among Malays

## Abstract

Malaysia is a multicultural and multiethnic country comprising numerous ethnic groups. From the total population of 32.7 million, Malays form the bulk of the Bumiputera in Malaysia comprise about 69.9%, followed by Chinese 22.8%, Indian 6.6%, and others 0.7%. The heterogeneous population and increasing numbers of non-citizens in this country affects the heterogeneity of genetic diseases, diversity, and heterogeneity of thalassaemia mutations. Alpha (α)-thalassaemia is an inherited haemoglobin disorder characterized by hypochromic microcytic anaemia caused by a quantitative reduction in the α-globin chain. A majority of the α-thalassaemia are caused by deletions in the α-globin gene cluster. Among Malays, the most common deletional alpha thalassaemia is −α^3.7^ deletion followed by −−^SEA^ deletion. We described the molecular characterization of a new −−^GB^ deletion in our population, involving both alpha genes in *cis*. Interestingly, we found that this mutation is unique among Malay ethnicities. It is important to diagnose this deletion because of the 25% risk of Hb Bart’s with hydrops fetalis in the offspring when in combination with another α^0^- thalassaemia allele. MLPA is a suitable method to detect unknown and uncommon deletions and to characterize those cases which remain unresolved after a standard diagnostic approach.

## 1. Introduction

Alpha (α)-thalassaemia is the most common inherited disorder resulting from a reduced rate of synthesis of the α-globin chain. The prevalence of alpha thalassaemia trait in local studies including the Southeast Asia regions is higher (4.08% to 48.9%) [1,2,3,4,5,6] compared to beta (β)-thalassaemia carriers, which are reported by many researchers and range from 0.93% to 12.8% [1,2,5,7]. In one meta-analysis study involving 83,864 thousand subjects in the Southeast Asian region, the prevalence of α-thalassaemia in Malaysia is 17.4%, which is lower compared to the overall prevalence in the Southeast Asian region, which is 22.6% [8]. The difference in frequencies could be attributed to the different populations studied, with different inclusion criteria and due to the relatively smaller sample size in particular studies. Genomic deletions involving the α-globin gene cluster on chromosome 16p13.3 are the most frequent causes. The function of the α-globin gene cluster 16p13.3 requires the presence of a major upstream regulatory element, referred to as the HS-40 because it is 40 kilobases (kb) upstream to the ζ2 locus [3]. To date, more than 70 different deletions involving the alpha globin gene clusters have been described (https://globin.bx.psu.edu/cgi-bin/hbvar/query_vars3 accessed on 1 March 2023).

Normal individuals have four alpha genes (αα/αα) per genome, two identical genes (αα) on each chromosome 16p. Structurally, the more telomeric α−globin gene is designated with the α2 gene (HBA2) and the more centromeric gene is designated with the α1 gene (HBA1) [6]. The α^0^-thalassaemia, involves complete loss of both functional α globin genes in *cis* on the same chromosome mainly caused by deletions (−−/αα). The molecular deletions resulting in α^0^-thalassaemia include the −−^SEA^, −−^THAI^, −−^MED^, and −−^FIL^ deletions and are presented with mild hypochromic microcytic anaemia [1,9]. The deletions vary in size and tend to be geographically isolated, with two particularly common ones in the Southeast Asia and Mediterranean region, which are −−^SEA^ and −−^MED^ deletions, respectively. Non-deletional α-thalassaemia can result from mutations of the α2 gene (α^T^α) or the α1 gene (αα^T^) [10,11]. Majority of cases with a non-deletional type of α-thalassaemia are much often caused by a mutation in the α2 gene than α1 gene due to the dominant effect of the α2 gene that leads to a more severe phenotype [2,3,12].

The severity of the defect is variable. Generally, α-thalassaemia syndrome has at least four different clinical manifestations. At one extreme is a completely asymptomatic condition, resulting from the deletion or dysfunction of one of the four α-genes, which produces either trivial abnormality in the blood count and film or silent carrier. The designated α^+^ (alpha plus) thalassaemia is used. At the other extreme there is a four gene deletion with a total lack of α-globin gene chain that causes haemoglobin Bart’s hydrops fetalis, a condition that is generally incompatible with life [9,13,14]. Haemoglobin H disease is a clinical syndrome resulting from a variety of genetic abnormalities. The most common causes are compound heterozygous abnormalities for α° and α^+^ thalassaemia, e.g., −−^SEA^/−α^3^.^7^ in South East Asia and −(α)^20^.^5/^−α^3^.^7^ or −−^MED^/−α^3^.^7^ in the Mediterranean area [3,10,15]. When both α-genes on a single chromosome are deleted, the designated α° thalassaemia (alpha zero thalassaemia) is used [9].

Malaysia is a multiethnic, multicultural, and multilingual society, and the many ethnic groups in Malaysia consist of a majority of Malays and other indigenous people including Bumiputera 69.9%, Chinese 22.8%, Indian 6.6%, and others 0.7%. According to current population estimates, Malaysia’s population comprised 32.7 million people in 2022 (Population and Housing Census, 2022 http://www.dosm.gov.my accessed on 1 March 2023). Malaysia has a heterogeneous population; thus, the distribution of thalassaemia carriers varies by geographical location and ethnic groups. The distribution of specific genotype determinants has been reported by several local researchers [7,8,9,10]. In Malaysia, most common alpha determinants that were reported are the −α^3^.^7^ (NG_000006.1:g.34164_37967del); −−^SEA^ (NG_000006.1: g.26264_45564del); and −α^4^.^2^ (NG_000006.1:g.30682_34939del) of the deletional type and α^CS^α (NG_000006.1:g.34461T>C) and rarely α^CD59^α (NG_000006.1:g. g.34071G>A) of the non-deletional type [1,10]. Among them, −−^SEA^ deletion is a more common genetic determinant among the Malaysian–Chinese followed by Malays [10,16,17]. The −α^3^.^7^ deletion is more commonly seen among Malays followed by the −−^SEA^ deletion and Hb Constant Spring [1,10,11]. Therefore, the fatal condition, Hb Bart’s hydrops fetalis, is reported mainly among Chinese, while Malays possess the single *α*-globin gene deletion that produces an asymptomatic disorder. Thalassaemia has rarely been observed in the Malaysian Indians who are migrants mainly from Southern India [10,18]. The mutation spectrum of thalassaemia is unique, with some diversity and peculiarity according to specific geographical location and distinctive ethnic groups [19,20]. In Malaysia, few alpha and beta variants have been reported as unique among Malays, namely α2 Codon 141 (CGT>CCT) Hb Singapore NG_000006.1:g.34459G>C, [20] and Codon 6 [GAG>GCG] Hb G-Makassar (NG_000007.3:g.70614A>C) [21]. In 2019, Malaysian Thalassaemia Registry reported a total of 8178 registered thalassaemia patients, of which majority of them suffered from HbE/β-thal; followed by β-thal major (β-TM); Hb H disease; β-thal intermedia (β-TI); and others, with the percentage of 35.19%, 32.66%, 19.48%, 9.02%, and 3.64%, respectively [22,23]. The ‘others’ group includes other forms of the HbH disease, Hb Lepore Hollandia, α-thalassaemia syndrome, δβ-thalassaemia, and other thalassaemia disorders requiring regular blood transfusions [23].

Conventional methods using either capillary electrophoresis (CE) and high-performance liquid chromatography (HPLC) are unable to definitively detect alpha thalassaemia carriers [11,24]. Although common deletions and non-deletional alpha genes relevant to the Southeast Asian population can be detected by a single multiplex Gap-PCR and ARMS-PCR, the rare and novel deletions depend on advanced techniques for their identification [9]. The multiplex ligation-dependent probe amplification (MLPA) technique has been used for this purpose and was successfully used in this study to detect the molecular alterations responsible for the α-thalassaemia phenotype in 29 individuals of Malay ethnicity that remained uncharacterized after sequential analysis and Gap-PCR for common deletions. This study aimed to describe uncharacterized deletion of −−^GB^ NG_000006.1:g.(13208_23659)_(38689_41963) in the Malaysian population with their genotypic interactions and to determine the haematological indices and molecular characteristics of this deletion.

## 2. Materials and Methods

### 2.1. Subject Recruitment

This was a retrospective DNA analysis of data of α-thalassaemia cases registered from 2012 to 2018. Analysis involved a total of 1054 subjects that were referred to our laboratory in the Institute for Medical Research (IMR), Malaysia, for further analysis of α-thalassaemia, in which no molecular defect has been found using common techniques. Molecular analysis for common deletions and mutations of alpha thalassaemia were completed in all samples using multiplex Gap-PCR method and Amplification Refractory Multiples System (ARMS) PCR method, respectively. The alleles tested using these platforms were able to detect seven types of deletions (−α^3.7^, −α^4.2^, −−^SEA^, −−^THAI^, −−^FIL^, −(α)^20.5^ and −−^MED^) [11,25,26,27] and six types of non-deletional mutations which include the initiation codon (ATG>A-G); codon 30 (∆GAG); codon 35 (TCC>CCC) Hb Évora; codon 59 (GGC>GAC) Hb Adana; codon 125 (CTG>CCG) Hb Quang Sze; and codon 142 (TAA>CAA) Hb Constant Spring [28].

From all samples analysed, 58 subjects that account for 5.5% of the sample were found to have unknown deletions with 16 different deletional types. Five deletions have been successfully characterized, namely −−^GB^, −−^AW^, −α^MAL3.5^, −(α)^4^.^9^ and −α^2^.^4kb^ deletions [24,29,30,31]. Another 11 deletional types remained uncharacterized at that time. Twenty-three (2.2%) subjects were found to have amplification of the alpha globin gene, namely ααα^anti−3^.^7^, ααα^anti−4^.^2^, and an uncharacterized amplification of the alpha globin cluster. Two-hundred-and-forty-nine (23.6%) subjects with variable findings such as gene conversion were recorded. Seven-hundred-and-twenty-four (68.7%) subjects were determined negative for any mutations or deletions based on α-MLPA and sequencing analysis. Twenty-nine subjects were found to have −−^GB^ deletions that account for 2.8% of the sample from total of this study’s population. The demographic data including age, race, gender, red blood cell indices, and Hb analysis findings were recorded.

### 2.2. Thalassaemia Screening Tests and DNA Analysis

The thalassaemia screening tests consisted of a full blood count (FBC) and Hb analysis. The FBC was completed using an automated haematology analyser. The Hb analysis was performed according to a set of tests, i.e., peripheral blood film; CE (SEBIA, Lisses, France); HPLC (Bio-Rad Laboratories, Hercules, CA, USA); and Hb electrophoresis (SEBIA, Lisses, France). Based on blood count data (MCV and MCH), all the cases were suspected to have α-thalassaemia; however, in the view of negative common genotypes by conventional methods, further analysis is required for α-thalassaemia genotyping. The cut-off for mean corpuscular haemoglobin (MCH) and mean corpuscular volume (MCV) in this study was <25 pg and <80 fl, respectively, based on the criteria for alpha genotyping in our centre.

Genomic DNA of the cases was extracted from peripheral blood samples using the QIAsymphony DSP DNA Kit (Qiagen, GmbH, Hilden, Germany) in accordance with the manufacturer’s instructions. All the samples were subjected to Multiplex Ligation-dependent Probe Amplification (MLPA) (Salsa MLPA P140-B4 lot 0415 HBA, MRC-Holland) and alpha sequencing analysis.

### 2.3. MLPA Reaction

To rule out uncommon deletion in the α-globin gene cluster, MLPA assay was performed in accordance with the manufacturer’s instructions. Thirty-four probes specific to the α- globin gene cluster including the regulatory element and multispecies conserving sequence 2 (MCS-R2; previously called HS-40) [11] were adjusted for detection of the copy number variation in the α-globin gene cluster. Threshold limits for deletion and duplication were set at <0.65 and >1.3, respectively. MLPA oligonucleotide probes upstream of the alpha pseudogene 1 (*HBAP1*), until the downstream of α1 globin gene (*HBA1*) (as shown in Figure 1), failed to ligate and amplify and hence had a significantly low ratio signal generated, indicating the presence of an uncharacterized deletion involving both α2 and α1 globin gene spanning from probe 8 to 27 (estimated size of the deletion was 16,771 bp). The MLPA findings were identical for all the cases (with the same deletional breakpoint).

### 2.4. Breakpoint Analysis

To confirm the deletion and to map the 5′ and 3′ breakpoint regions, a Gap-PCR method was developed, and its amplicon was sequenced. Breakpoint areas are defined as the region between the position of the last MLPA probe present and the first probe deleted. A series of Gap-PCR primers were designed to flank the deleted region observed on the MLPA. The optimal and reproducible amplification entrapping the deletion region was successfully achieved with the forward PCR primer GB Fw (5′-ACCCACACTGAGCCTCAAACAG-3′) and reverse primer GB Rev (5′-TACCATTGTAGCCATTTTTCAGGT-3′). The targeted amplicon was 1481 bp in size. The amplification was carried out in a final volume of 50 µL containing 0.1 mg genomic DNA, HotStarTaq^®^ Plus Master Mix (2.5 units HotStarTaq DNA Polymerase, 1X PCR Buffer, 1.5 mm MgCl_2_ and 200 µM of each dNTP) (Qiagen GmbH, Hilden, Germany); 0.5X Q-solution; and 1.2 µM of primer GB Fw and GB Rev each. PCR amplification was carried out using Eppendorf Mastercycler^®^ ProS (Eppendorf AG, Hamburg, Germany) with an initial denaturation at 95 °C for 5 min followed by 30 cycles of amplification (denaturation at 98 °C for 45 s, annealing at 64.0 °C for 90 s, elongation at 72 °C for 135 s) and final extension at 72 °C for 5 min. The amplicon was checked using gel electrophoresis (1.2% *w*/*v* agarose gel) for 60 min at 110 voltages. Cycle sequencing of the purified PCR products were then performed using the BigDye^®^ Terminator v3.1 cycle sequencing kit and the sequences were studied using the ABI 3730XL DNA Analyser (Applied Biosystems, Foster City, CA, USA). The sequencing data were analysed using CLC Main Workbench (CLC Bio, Aarhus, Denmark) (Figure 2).

The α-globin gene sequencing was carried out in all 29 cases for *HBA1* and *HBA2* gene ranging from −50 to 5′ of the untranslated region (UTR) until the Poly A conserved region to detect any presence of α-thalassaemia mutation in both genes. The alpha sequencing results were normal in 25 cases; a total of 2 cases with the Hb Constant Spring (Hb CS) NG_000006.1: g.34461T>C, 1 with IVS II-142 (G>A) and 1 with IVS II-55 variation were found. Figure 3 show the gap-PCR assay for −−^GB^ deletion.

### 2.5. Statistical Analysis

The demographic data including the age and ethnicity of the −−^GB^ deletion were analysed with descriptive analysis. All statistical analyses were performed using SPSS software (Ver. 22, SPSS Inc., Chicago, IL, USA).

### 2.6. Ethical Approval

This study was conducted in accordance with the Declaration of Helsinki and approved by the National Medical Research Register, the regional ethical board of Malaysia. Written informed consent for molecular genotyping was obtained from the subjects prior to blood taking.

## 3. Results

All 29 −−^GB^ deletion cases were of Malay ethnicity except one non-citizen case from Indonesia. From a total of 29 cases with the −−^GB^ deletion, 23 cases (79.3%) were heterozygous alpha zero deletion (based on phenotype and genotype findings), while 6 cases (20.7%) possessed the non-deletional alpha thalassaemia (Hb CS, IVS II-142) and deletional type (−α^4.2^). Their age ranged from 3 to 37 years for heterozygous −−^GB^ deletion. There were (*n* = 8/23, 34.7%) males and (*n* = 15/23, 65.2%) females. No specific geographical distribution unique for this deletion was observed; however, Kedah and Negeri Sembilan that are located in the Northern and Southern parts of Malaysia, respectively, have the highest incidence.

### 3.1. Haematological Characteristics

An overview of the demographic and haematological parameters of 23 cases of heterozygous −−^GB^ deletion is given in Table 1. Most of the cases presented with normal or mildly low haemoglobin (Hb) levels in heterozygous genes with a mean Hb level of 13.1 g/dL for males and 10.32 g/dL for females. The means for MCV, MCH, and HbA2 using HPLC and CE were 65.4 fL, 20.6 pg, 2.3%, and 2.2%, respectively. While the Hb F level for this alpha zero deletion was mildly raised at 1.2%, six cases were reported as compound heterozygous with the −−^GB^ deletion with either a mutation or a single alpha gene deletion, namely the Hb CS (NG_000006.1g.34461T>C); IVII-142(G>A) (NG_000006.1:g.34334G>A); and −α^4.2^ deletion (NG_000006.1g.30682_34939del) (Table 2). We also reported, for the first time in the Malaysian population, a rare case of the Hb H disease with a combination of heterozygous −−^GB^ deletion and IVS II-142 (G>A). The patients possessing the Hb H disease at an older age had haemoglobin levels ranging from 6.4 to 9 g/dL along with their Hb levels dropping during infection. This is a novel variant affected by the splice acceptor site of the α2-globin gene (consensus sequence of AG to AA) that leads to complete suppression of the α-globin gene, which was reported in the Argentinian family with the Hb H disease. The patient at the age of 67 years with jaundice and hepatomegaly and a Hb of 6.6 g/dL required 10 transfusions to correct her anaemia [12]. All other cases possessed thalassaemia intermedia except for one case who had severe anaemia at 4 years old (case 2, Table 2) with a Hb level of 4 g/dL after the course of infection. However, the patient currently required 1–2 transfusions in a year with no complications in iron overload. He is currently treated with a folic acid tablet. As expected in compound heterozygous −−^GB^ deletion, the mean Hb A2 is lower when compared to the heterozygous cases with the levels of 1.9% and 2.3%, respectively. In contrast, the Hb F levels for compound heterozygotes cases are higher than heterozygotes cases with the levels of 2.1% and 1.2%, respectively.

### 3.2. MLPA and Breakpoint Analysis

The MLPA findings were identical for all 29 cases (with same deletional breakpoint) ranging from probes upstream of the alpha pseudogene 1 (*HBAP1*) until the downstream of a1 globin gene (*HBA1*) (as shown in Figure 1) failed to ligate and amplify; hence, it had a significantly low ratio signal generated, indicating the presence of an uncharacterized deletion involving both the a2 and a1 globin gene spanning from probe 8 to 27 (estimated size of deletion was 16,771 bp). The exact breakpoint of this deletion was determined by Gap-PCR and direct sequencing (described in the methodology). A similar deletion was previously described in a Dutch individual of mixed ethnicity, Indonesian and Arabic [24,32,33], respectively.

## 4. Discussion

In the Institute for Medical Research (IMR), most cases sent to the laboratory for further analysis of α- thalassaemia after exclusion of common deletions and mutations of alpha thalassaemia, namely −−α^3.7^, −α^4^.^2^, −−^SEA^, −−^THAI^, −−^FIL^, −−^MED^, −(α)^20.5^, and non-deletional mutations which include the initiation codon (ATG>A-G); codon 30 (∆GAG); codon 35 (TCC>CCC) Hb Évora; codon 59 (GGC>GAC) Hb Adana; codon 125 (CTG>CCG) Hb Quang Sze; and codon 142 (TAA>CAA) Hb Constant Spring. Currently, we are using α-MLPA and sequencing platforms for further analysis of α-thalassaemia. From the analysis of a total of 1054 patients from 2012 to 2018 we found 29 cases with the −−^GB^ deletion with a frequency of 2.8%. The deletion breakpoint was successfully characterized using MLPA, based on the breakpoint characterization with a set of Gap-PCR and sequencing methods described previously [24,32,33].

The −−^GB^ deletion was successfully characterized in a Dutch individual of Arabic and Indonesian ethnicity, based on a previous report by [24,32,33]. Recently, Hottentot et al. raised the issues regarding an inaccurate deletion annotation as the −−^GB^ deletion with a 15.2 kb deletion by Mota et al., with correspondence to the IthaID: 3294, and without determining the exact breakpoint [33,34]. Determination of precise breakpoints is important before the deletion annotation and to prevent duplication registration pertaining of similar deletion that lead to inaccuracy of the databases [33]. Thus, we confirmed the findings of the −−^GB^ deletion with a 16,771 kb deletion in our population by designing a set of Gap-PCR primers to define the breakpoint characterization by the MLPA followed by confirmation of the breakpoint with direct sequencing methods.

The demographic and haematological parameters of the 23 subjects with the heterozygous −−^GB^ gene deletion are given in Table 1. Most of the cases had normal or mildly low haemoglobin (Hb) levels in the heterozygous gene with a mean Hb level of 13.1 g/dL for males and 10.32 g/dL for females. Means for MCV and MCH were 65.4 fL and 20.6 pg, respectively. These results were consistent with the other two alpha gene deletions, namely the −−^SEA^, −−^THAI^, and −−^FIL^ deletion [6,10,17]. The two parameters of MCV and MCH may give a significant predictive clue that helps distinguish the type of deletional alpha thalassaemia. The deletion of 16.7 kb −−^GB^ involved both alpha globin genes as well as embryonic zeta (ζ), and pseudogenes. In our population, the −−^SEA^ deletion was the most common genetic determinant found among the Malaysian-Chinese population, based on a few local studies [1,3,10]. However, in Malays, the most common alpha determinants were the alpha plus thalassaemia (−α^3^.^7^) followed by the alpha zero (−−^SEA^) deletion and the αα^CS^ with an allele frequency of 13.4%, 9.2%, and 4.2%, respectively [10]. Considering Malaysia as a wide representative of multiple ethnic groups in the Peninsular and West Malaysia, the −−^SEA^ deletion was found to be diverse across ethnicity. The relative chromosome frequency of the −−^SEA^ deletion was 0.70% in the Chinese population, followed by Malays (0.16%), and Sabah and Indian population with 0.06% and 0.043%, respectively [6]. A study from R. Ahmad et al. reported higher allele frequency of the −−^SEA^ deletion among the Chinese population which was 0.86%, followed by Malays with 0.46% [10]. The difference could be explained due to studying different populations with different sample sizes involved. Interestingly, all the cases of the −−^GB^ deletion reported in our population were of Malay ethnicity, except for one case, which was an immigrant from Indonesia (accounts for about 2.8% incidence). All individuals with the −−^GB^ deletion were not known to be related. We proposed that this deletion is a unique finding among Malays in our population and could be the most common alpha zero determinant among them. This finding should correspond with the larger scale of study. In terms of gene mapping, the deletion is distributed evenly in Malaysia, but the incidence is higher in the Northern Region, such as in the Kedah Region, and in the Southern Region, such as the Negeri Sembilan. Kedah is located on the border of Northern Thailand where the incidence of alpha thalassaemia, particularly the −−^THAI^ and −−^SEA^ deletion, is higher [15]. The incidence of the −−^GB^ deletion could be under-reported due to the test limitations.

The real incidence of this deletion could not be predicted since this deletion is not included in the common deletional type of alpha thalassaemia by conventional multiplex Gap-PCR. However, based on a retrospective analysis of the 6 years of data from our study, (after exclusion of common deletional and non-deletional alpha thalassaemia) the incidence of the −−^GB^ deletion is about 2.8% among our populations and could be the second most common α°-deletion in the Malaysian population after the −−^SEA^ deletion.

The −α^3.7^ deletion is the most common alpha genotype among Malays reported by various researchers, followed by −−^SEA^ deletion [1,10,16]. Whether the −−^GB^ deletion was under-reported should be investigated further, especially among Malays that are negative for the common deletional alpha thalassaemia panels routinely performed in our laboratory.

The mean MCV value for the −−^GB^ deletion is similar to other double deletional types of alpha thalassaemia, and was 65.4 ± 3.58, almost in the range that was reported by [13,19]. Among the seven most common deletional and non-deletional alpha thalassaemia, namely −^3.7^, −−^SEA^, −−^FIL^, −−^THAI^, −(α) ^20.5^, Hb Quong Sze, Hb Constant Spring, CD59, and the −−^GB^ deletion was the only deletional type of alpha thalassaemia that were found to be mutually exclusive among the Malay population in our country. 

In Malaysia, the Hb H disease is common among the Chinese population (34.21%) in comparison to Malays (21.54%) and Indians (13.89%) (data from Annual Report of Malaysian National Thalassaemia Registry, 2019). The discovery of this deletion in routine testing may affect the incidence of the Hb-H disease among Malays. The Chinese population had a higher incidence of the Hb H disease as they had the highest incidence of the alpha zero −−^SEA^ deletion [6,9].

### Limitation

This study had a few limitations. The incidence reported here would not precisely correspond to the populations’ prevalence of the −−^GB^ deletion as cases that were negative for common deletions and non-deletional α-thalassaemia, that required further analysis, were counted as a denominator without considering the status of iron. The value of the red cell parameters in this study, especially MCV, MCH, and RDW, could be affected by co-inheritance with iron deficiency anaemia, which was not thoroughly investigated in this study.

## 5. Conclusions

We described the molecular characterization of a new −−^GB^ deletion, involving both alpha genes in *cis*. Interestingly, we found that this mutation presents unique findings in the Malay ethnicity in our country, whereas another study has reported this deletion among a Dutch individual of Arabic and Indonesian ethnicity. It is important to diagnose this deletion because of the 25% risk of Hb Bart’s with hydrops fetalis in the offspring when in combination with another α^0^ -thalassaemia allele. The MLPA is a suitable method to detect unknown and uncommon deletions and to characterize those cases which remain unresolved after a standard diagnostic approach.

## Figures and Tables

**Figure 1 diagnostics-13-03286-f001:**
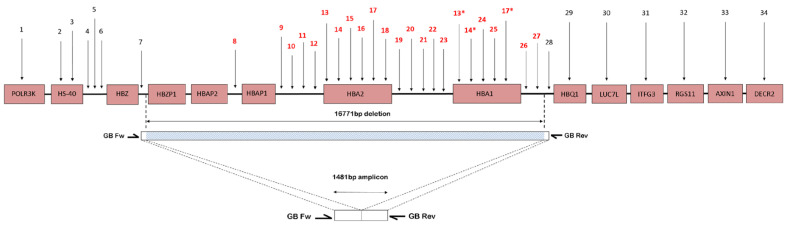
Schematic presentation of part of the short arm of chromosome 16. The pink boxes represent the genes. Numbers refer to MLPA-probe pairs distribution along the cluster and surrounding regions according to Harteveld et al. (2005) [32]. The blue bar indicates the deletion length and position, and white area bars represent the area where the breakpoint is located. The MLPA probes were scattered along the α-globin gene cluster. The primers (GB Fw and GB Rev) were used to amplify the sequences across the deletion breakpoint. The amplicon with size of 1481 bp was sent for direct sequencing to identify the deletion breakpoint. The deletion was 16,771 bp in size, which deletes both *HBA2* and *HBA1* genes. * Indicates that the probes detect sequences present in both *HBA1* and *HBA2*.

**Figure 2 diagnostics-13-03286-f002:**
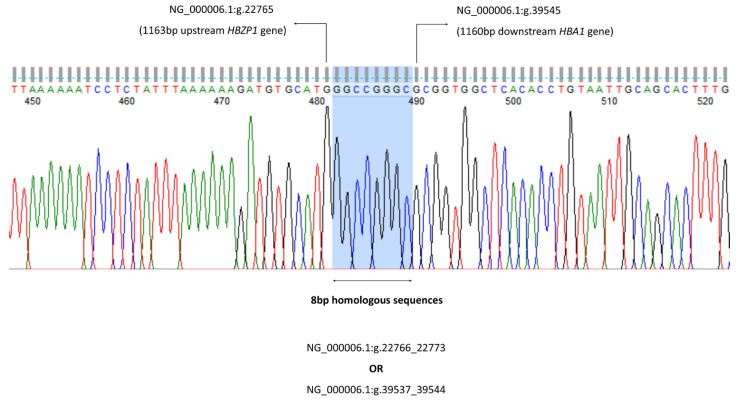
The 8 bp homologous sequences can be found at both 5′ and 3′ deletion breakpoints either at NG_000006.1:g.22766_22773 or NG_000006.1:g.39537_39544. Thus, the −−^GB^ deletion can be mapped either from the 1164 bp upstream of *HBZP1* to 1151 bp downstream of *HBA1* gene (NG_000006.1:g.22766_39536del), or from the 1156 bp upstream of *HBZP1* to 1159 bp downstream of *HBA1* gene (NG_000006.1:g.22774_39544del).

**Figure 3 diagnostics-13-03286-f003:**
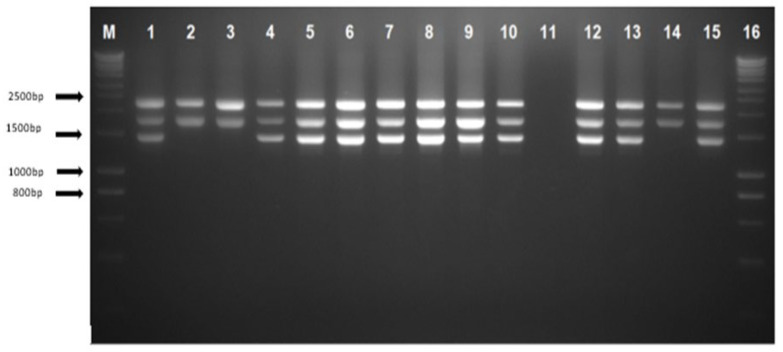
Agarose gel electrophoresis of PCR products using gap-PCR assay. Lanes M and 16 show DNA marker ladders (hyperladders). Lane 1 shows positive control for heterozygous --^GB^ deletion; lane 2 shows normal control with the 2503 bp depicting the LIS internal control band and the 1800 bp α2 globin gene band; lanes 4–10, 12–13, and 15 indicate heterozygous --^GB^ deletion; lane 11 shows a negative control template; and lanes 3 and 14 show samples with a normal result.

**Table 1 diagnostics-13-03286-t001:** Patients’ demographic, haematological parameters, and Hb profiles for heterozygous −−^GB^ deletion.

No	Gender	Age	Hb (g/dL)	RBCs (10^6^/µL)	RDW (%)	MCV (fL)	MCH (pg)	HbA_2_ (%)	HbF (%)	Genotype
1	M	16	14.7	6.8	13.2	66.8	21.7	2.5 ^a^	<1.0 ^a^	−−^GB^/αα
2	M	31	13.5	6.6	15.2	68.7	20.3	2.0 ^a^	-	−−^GB^/αα
3	F	33	11.5	5.4	17.2	67.0	21.2	2.2	0.4	−−^GB^/αα
4	M	16	15.2	7.4	19.1	60.1	20.5	2.5 ^a^	0.6 ^a^	−−^GB^/αα
5	F	22	10.4	4.7	16.7	70.0	22.0	2.2	0.5	−−^GB^/αα
6	M	16	12.0	6.0	17.7	62.1	19.9	2.6 ^a^	-	−−^GB^/αα
7	M	17	13.1	5.7	13.3	68.0	23.0	2.9	0.3	−−^GB^/αα
8	M	28	12.0	5.3	14.7	68.5	22.6	2.3 ^a^	-	−−^GB^/αα
9	F	37	11.4	5.6	15.1	65.2	20.5	3.1	0.6	−−^GB^/αα
10	F	36	12.3	6.3	15.9	62.1	19.5	2.4	0.2	−−^GB^/αα
11	F	31	11.1	5.6	17.6	60.8	19.9	2.4	0.2	−−^GB^/αα
12	F	37	11.5	5.7	18.2	63.7	20.1	2.4	0.2	−−^GB^/αα
13	F	30	11.3	5.7	-	63.3	19.6	2.8	0.2	−−^GB^/αα
14	F	27	11.1	5.4	16.7	66.0	20.5	2.2	0.1	−−^GB^/αα
15	F	4	10.6	5.4	14.8	61.0	19.7	2.5	0.6	−−^GB^/αα
16	F	33	10.7	5.4	18.9	61.1	19.7	2.3 ^a^	-	−−^GB^/αα
17	F	3	11.6	6.1	37.9	63.9	19.0	2.1	8.1	−−^GB^/αα
18	F	28	10.8	5.4	14.6	71.5	20.1	2.1	1.1	−−^GB^/αα
19	F	34	11.4	5.7	-	62.2	19.9	2.2	1.2	−−^GB^/αα
20	F	7	-	-	14.7	-	-	2.1	-	−−^GB^/αα
21	M	5	12.0	5.3	20.1	68.5	22.6	2.5 ^a^	0.5 ^a^	−−^GB^/αα
22	M	36	12.9	6.0	-	71.7	21.4	1.9	1.3	−−^GB^/αα
23	F	28	10.1	5.0	27.4	66.4	20.3	1.0 ^a^	-	−−^GB^/αα
Mean		24.1	11.9	5.8	17.5	65.4	20.6	2.3	1.2	
								2.2 ^a^	0.6 ^a^	
(SD)		(11.31)	(1.31)	(0.62)	(5.33)	(3.58)	(1.12)	(0.49)	(1.98)	
								(0.52) ^a^	(0.32) ^a^	

HbA2 and HbF results are based on high-performance liquid chromatography results except for ^a^, which are based on capillary electrophoresis results. ‘-‘ data not available in the record.

**Table 2 diagnostics-13-03286-t002:** Patients’ demographic and clinical data for compound heterozygous −^GB^ deletion.

No	Gender	Age	Hb (g/dL)	RBCs (10^6^/µL)	MCV (fL)	MCH (pg)	HbA_2_ (%)	HbF (%)	HbH (%)	HbCS, Bart	Genotype
1	F	3	8.7	5.3	62.7	16.3	0.5 ^a^	0.4 ^a^	5.8 ^a^	2.7 ^a^, 0.6 ^a^	−−^GB^/α^HbCS^α
2	F	4	4.3	2.5	70.6	17.6	1.1 ^a^	3.1 ^a^	1.2 ^a^	1.2 ^a^, 0.8 ^a^	−−^GB^/α^HbCS^α
3	F	7	9.8	6.2	54.8	15.8	1.5	2.3	pre run peak no abnormal peak pre run peak no abnormal peak		−−^GB^/-α^4.2^
4	F	10	8.9	5.0	65.9	17.9	1.4	3.0			−−^GB^/-α^4.2^
5	M	62	6.4	3.2	74.1	20.0	1.2	<1.0			−−^GB^/α^IVSII−142(G>A)^α
6	M	1	9.9	5.4	63.7	18.3	2.6	2.7			−−^GB^/α^IVSII−55 var^α
	Mean	14.5	8.0	4.6	65.3	17.7	1.9	2.1	3.5		
	(SD)	(23.5)	(2.2)	(1.42)	(6.7)	(1.5)	(0.7)	(1.1)	(3.25)		

Abbreviations: RBC, red blood cells; Hb, haemoglobin; MCV, mean cell volume; MCH, mean cell haemoglobin; MCHC, mean cell haemoglobin concentration; RDW, red cell distribution width; CE, capillary electrophoresis; HPLC, high performance liquid chromatography. The results are based on high-performance liquid chromatography results except for ^a^, which are based on capillary electrophoresis results. Pre run peak is a sharp peak, which occurs before the start of integration, indicating a HbH peak in chromatogram (HPLC) in the first minute of elution.

## Data Availability

The data presented in this study are available upon request from the corresponding author.

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
