# Peer review of "Characterization of New Alpha Zero (α^0^) Thalassaemia Deletion (−−^GB^) among Malays in Malaysian Population"

_diagnostics, 2023, doi:10.3390/diagnostics13203286_

Round 1

Reviewer 1 Report

The manuscript entitled "Characterization of New Alpha Zero (α0) Thalassaemia Deletion (--Gb) Among Malays in Malaysian Population" has been reviewed.

This is a retrospective study including α-thalassaemia cases.

This manuscript will give new information for specliasts related with α-thalassaemia.

Author Response

Thanks for the comments.

Reviewer 2 Report

The authors are to be commended for a nicely done and described study of the methods they used to identify and analyze individuals with a deletion which is fairly newly described a deletion of two genes in cis called--GB. The  authors describe the demographics of alpha thalassemia in their country and the importance of characterizing this mutation since it was a considerable number of patients with unknown mutations and these individuals can have offspring with Hydrops fetalis.

I have a few suggestions for modifications (very few).

1. There are many parts of the text which are repetitious. This makes some parts boring. The worst is in the discussion in which the whole first half is repetitious of the methods and results. 

2. Lines 83-84 and other parts of the manuscript: the authors use only the common terminoogy and not the proper genetic terminology (which is indeed used on lines 94-96). Both terminologies are usually used together. FOr brevity the authors can state they will use common names in the majority of the manuscript but at least on first mention the correct nomenclature should be used.

3. Line 117: Instead of "kept" say "registered" or accumulated. Also the word "incident" is used several times and should be "incidence". There are some other minor errors thoughout the text.

4. Line 199 in the figure legend there are some grammatical errors ("pink box represent gene") do they mean the pink boxes represent the genes?

5. I think Figure 3 is unnecessary we all know what a PCR agarose gel looks like.

6. lines 243 says the HbF level in the heterozygotes was 1.2% but on line 258-9 the authors say that the HbF level of the compound heterozygotes was lower as expected than heterozygotes and that was 2.1%. 2.1 is higher than 1.2. I may have missed something here.

7. Table 1 the MCV is a bit lower than what we see in my country with alpha thal zero heterzygosity could the patients also have iron deficiency? In particular since the RDW was high (17.5). Ranges of normal should be given at the bottom of the table.

8. Table 2 I am not sure what a pre run peak is. Also patient 6 has no result for HbCS or Barts, was it not done?

9. The individuals found to carry the --GS were all of Malay ethnicity, please state if they are not known to be related.

As above.

Author Response

2,3,4,6-Thank you for the comments. I have made the changes according to the suggestion and highlighted in yellow

5-We plan to keep Figure 3 because the figure represents the complete steps in molecular characterization of --GB deletion (MLPA, sequencing, and Gap PCR for confirmation) and we think it helped the reader to understand the methodology.

6- Thank you for highlighting this. I have made the changes. 

7-Regarding RDW, yes we already included in our limitation of the study and this study analyzed data from various labs from all over Malaysia (we are central lab), and the normal ranges could be slightly different across laboratories. However, I added information on red cell parameters in the limitation 

8- I made some statements on pre run peak in the revised manuscript. HPLC or CE were done in all the patients, however, the peak was not detected in case 6

9- Changes made as suggested 

Reviewer 3 Report

The Authors present the result of a retrospective analysis on a group of 1054 subjects sent to their laboratory for further analysis of alpha-thalassemia because no molecular defects were found with the common technique. The analysis was based on a newly identified DNA mutation affecting the alpha-thalassemia spectrum. Furthermore, the molecular characterization of a novel deletion --GB 21 in our population, involving both alpha in cis genes and the implication on the risk to determine an Hb Bart (Hydrops foetalis) has been reported.

The text reports the term "intermediate" replaced in recent years with the term "non-transfusion dependent thalassemia" which includes the group of patients who can live with lower than normal levels and who can only occasionally be transfused with packed red blood cells .

Minor comments

line 24. replace foetalis with fetalis

from line 33 to line 43 Very confusing data are reported because it all refers to the southeast asia region but with very different percentages. Everything should be rewritten, better specifying which regions have percentages of up to 80% of carriers and making the message you want to give to the reader clearer.

line 54 At the other extreme there is ..........

line 59 .........Mediterranean area

from line 62 to line 74 This paragraph could be moved to line 49 before the paragraph "the severity of defects ........."

line 100 correct Thalassemia

from line 128 to line 139 Results: the study examined 1054 subjects (58+23+249+724+29) but the sum is 1083. It is necessary to clarify better or report everything in a graph.

line 143 and line 144. Is the CE method (SEBIA, France) the same as Hb electrophoresis (SEBIA, France)?

line 144: On the basis of blood count data (MCV and MCH), ....

line 148: what was the limit value for MCV?

line 331 and 336 replace incident with incidence

line 323 insert the first author's name instead of the bibliography number

line 344 insert the name of the first author for each of the two works instead of the bibliography number

a revision of the text is necessary for several spelling errors or errors in the wording of the units of measurement. For example in line 240 13.1/dL instead of 13.1 g/dL, or line 255 4g/dL instead of 4 g/dL

line 268 insert the name of the first author for each of the two works in place of the bibliography number

line 284 check the sentence

line 296 and 299 insert the name of the first author for each of the two works in place of the bibliography number. Check for other similar situations throughout the text.

line 299 check the sentence

Bibliography

line 82 the work Haemoglobinopathies in southeast Asia is missing

S, Fucharoen, P Winichagoon - The Indian journal of medical Research 2011 Oct; 134(4): 498–506 - ncbi.nlm.nih.gov

Proofreading by a native speaker is required

Author Response

1-Thank you for the comments. In this manuscript, I'm using British English so I've changed it accordingly.

In some prevalent areas, a carrier frequency of mild form of (α+) reaches 80% or more and more severe forms (α°) reach their highest frequency in the Southeast Asia, where the carrier frequency can reach up to 10% [8]. I omit this statement as it may cause confusion to reader.

A few corrections are highlighted in blue (in the manuscript) as suggested by the reviewer

Is the CE method (SEBIA, France) the same as Hb electrophoresis (SEBIA, France)? Yes, it similar both are from SEBIA because the principle of the test is almost similar

From line 62 to line 74 This paragraph could be moved to line 49 before the paragraph "the severity of defects ........." (changes made and highlighted)

from line 128 to line 139 Results: the study examined 1054 subjects (58+23+249+724+29) but the sum is 1083. Answer: A total of 1054 subjects were examined. 58 subjects were found to have unknown deletions with 16 different deletional types. 249 subjects with variable findings such as gene conversion. 23 subjects were found to have amplification of alpha globin gene namely αααanti-3.7, αααanti-4.2 and 724 subjects were negative for any mutations or deletions based on α-MLPA and sequencing analysis. The sum is 1054. 29 cases of GB deletion are summed under a total of 58 samples positive with unknown deletion. 

line 144: On the basis of blood count data (MCV and MCH), correction done

line 148: what was the limit value for MCV? The value was added in the manuscript

line 323 insert the first author's name instead of the bibliography number and line 344 insert the name of the first author for each of the two works instead of the bibliography number- revision done throughout the manuscript. 

Reg : A revision of the text is necessary for several spelling errors or errors in the wording of the units of measurement. For example in line 240 13.1/dL instead of 13.1 g/dL, or line 255 4g/dL instead of 4 g/dL : A revision on the unit done